# Ethyl Hydroxyethyl Cellulose—A Biocompatible Polymer Carrier in Blood

**DOI:** 10.3390/ijms23126432

**Published:** 2022-06-09

**Authors:** Anja Eckelt, Franziska Wichmann, Franziska Bayer, John Eckelt, Jonathan Groß, Till Opatz, Kerstin Jurk, Christoph Reinhardt, Klytaimnistra Kiouptsi

**Affiliations:** 1Center for Thrombosis and Hemostasis (CTH), University Medical Center of the Johannes Gutenberg-University Mainz, Langenbeckstrasse 1, 55131 Mainz, Germany; eckelta@uni-mainz.de (A.E.); fwichman@students.uni-mainz.de (F.W.); franziska.bayer@kgu.de (F.B.); kerstin.jurk@unimedizin-mainz.de (K.J.); christoph.reinhardt@unimedizin-mainz.de (C.R.); 2WEE Solve GmbH, Auf der Burg 6, 55130 Mainz, Germany; john.eckelt@wee-solve.de; 3Department of Chemistry, Johannes Gutenberg University, 55099 Mainz, Germany; jgross03@uni-mainz.de (J.G.); opatz@uni-mainz.de (T.O.); 4German Center for Cardiovascular Research (DZHK), University Medical Center of the Johannes Gutenberg-University, Mainz Parter Site Rhine-Main, Langenbeckstrasse 1, 55131 Mainz, Germany

**Keywords:** polymer, nanomaterial, ethyl hydroxyethyl cellulose, platelets, plasma expanders

## Abstract

The biocompatibility of carrier nanomaterials in blood is largely hampered by their activating or inhibiting role on the clotting system, which in many cases prevents safe intravascular application. Here, we characterized an aqueous colloidal ethyl hydroxyethyl cellulose (EHEC) solution and tested its effect on ex vivo clot formation, platelet aggregation, and activation by thromboelastometry, aggregometry, and flow cytometry. We compared the impact of EHEC solution on platelet aggregation with biocompatible materials used in transfusion medicine (the plasma expanders gelatin polysuccinate and hydroxyethyl starch). We demonstrate that the EHEC solution, in contrast to commercial products exhibiting Newtonian flow behavior, resembles the shear-thinning behavior of human blood. Similar to established nanomaterials that are considered biocompatible when added to blood, the EHEC exposure of resting platelets in platelet-rich plasma does not enhance tissue thromboplastin- or ellagic acid-induced blood clotting, or platelet aggregation or activation, as measured by integrin α_IIb_β_3_ activation and P-selectin exposure. Furthermore, the addition of EHEC solution to adenosine diphosphate (ADP)-stimulated platelet-rich plasma does not affect the platelet aggregation induced by this agonist. Overall, our results suggest that EHEC may be suitable as a biocompatible carrier material in blood circulation and for applications in flow-dependent diagnostics.

## 1. Introduction

Nanomaterials can be functionalized to serve as nanocarriers or biomodulators for therapeutic applications [1,2]. One possible application is their use in targeted cancer therapy [3,4]. In addition to the non-toxic nature of nanomaterials, an essential requirement for the safety of nanomaterial therapeutics in human blood circulation is that they do not change the rheological properties of blood plasma and neither activate nor inhibit the clotting system. For example, amphiphilic macromolecule assemblies consisting of a hydrophilic polyethylene glycol tail and a branched hydrophobic head suppress platelet adhesion, a property favorable in the application in drug-eluting stents [5]. In contrast, silica nanomaterials, depending on their protein corona, have the capacity to support platelet activation [6,7].

The role of nanoparticles in the blood circulation is strongly dependent on their surface properties and the chemistry of the nanomaterial [8]. Silica nanoparticles, for example, disturb the vascular barrier, and are therefore unsuitable for intravascular applications. They down-regulate endothelial junction proteins, thus disturbing the blood–brain barrier [9]. Furthermore, they disrupt VE–cadherin interactions, inducing endothelial leakages that support cancer extravasation and metastasis [10]. Silica nanoparticles interact with red blood cells [11] and have been found to promote platelet microaggregation [12,13,14]. Although the interactions of silica nanoparticles with blood cells are largely unfavorable, the effects of other nanomaterials on human platelets remain ill-defined.

The interference of nanomaterials with platelets and the resulting enhancement of platelet aggregation behavior may strongly hamper their use in therapeutic applications, blood analytics, and diagnostics [15]. The use of nanomaterials as carriers in the bloodstream requires that these materials minimally interfere with the clotting system. This requirement is, for instance, relatively well met by plasma expanders such as gelatin polysuccinate (Gelafusal^®^) and hydroxyethyl starch (HES) (Volulyte^®^ or Vitafusal^®^), which are routinely used in clinics to increase oncotic pressure to prevent shock caused by severe blood loss [16]. In particular, gelatin solutions (Gelafusal^®^) have been shown to have minimal effects on blood clotting in whole blood thromboelastography analyses [17]. In addition, studies on platelet activation have demonstrated that HES affects platelet activation or aggregation at resting and activating conditions [18,19,20]. However, to functionalize nanomaterials as carriers, new nanomaterials are needed that ideally reflect the rheological properties of human blood without activating the hemostatic system [21]. Our goal is to examine whether ethyl hydroxyethyl cellulose affects ex vivo clot formation and platelet function, thus considering it as a potential candidate to substitute the plasma expanders currently in clinical use.

## 2. Results

Comparing ethyl hydroxyethyl cellulose (EHEC) with other biopolymers with application in the blood circulation (Gelafusal^®^, Volulyte^®^, and Vitafusal^®^) (Table 1), we here propose the use of EHEC as a biocompatible material that does not interfere with the clotting system while resembling the rheological properties of human blood. The aim of our study was to investigate whether EHEC solutions have a blood-like rheology profile and whether they affect clot formation and platelet aggregation behavior. This is critical because any effect on those parameters will likely preclude their use as a biocompatible carrier material for future intravascular therapeutic applications.

Polymers are characterized by their molar mass distribution. The molar mass distribution of EHEC, quantified by size exclusion chromatography, is shown in Figure 1A. The viscosity of human blood exhibits a shear-thinning behavior, i.e., the viscosity decreases with increasing shear rate (Figure 1B). This property can be well mimicked by EHEC solutions. In contrast, established blood plasma expanders exhibit Newtonian behavior, i.e., constant viscosity with changing shear rates. Consequently, the velocity of EHEC solutions is comparable to that of human blood, even if the diameter of the blood vessel changes.

Rotational thromboelastometry (ROTEM) is widely used to rapidly assess the viscoelastic changes that occur during clotting and is an essential part of patient blood management [22]. Here, we used two different tests to trigger coagulation: EXTEM, in which coagulation is initiated by a reagent containing tissue thromboplastin (tissue factor), and INTEM, in which clot formation is triggered by ellagic acid. Whole blood with different concentrations of EHEC or the vehicle (NaCl) (2.5%, 5%, and 10%) did not enhance tissue thromboplastin-induced clot formation (Figure 1C–F). The supplementation of whole blood with 10% EHEC even resulted in prolonged clot formation time when compared with either whole blood or whole blood with the vehicle (Figure 1D). Similar observations were made for the initiation of clotting by the contact phase (INTEM) (Figure 1G–J). Thus, supplementation with EHEC does not enhance clot formation in whole blood but rather prolongs clotting at higher concentrations.

Since the exposure of human platelet-rich plasma (PRP) to certain nanomaterials is known to result in quantitative platelet aggregation, we exposed EHEC solution (0.38% *w*/*w*) to PRP and performed light transmission aggregometry experiments. Because the addition of EHEC was comparable to the addition of physiological NaCl solution (0.9% *w*/*w*), we did not detect a significant increase in platelet aggregation under resting conditions (Figure 2A).

Integrin α_IIb_β_3_ is the major platelet receptor responsible for fibrinogen bridge formation and subsequent platelet aggregation. Platelet activation induces a conformational change in integrin α_IIb_β_3_, which can be detected by flow cytometric analysis using the antibody clone PAC-1 [23]. To test whether exposure to EHEC triggers platelet activation, we analyzed the activation state of α_IIb_β_3_ and the cell surface exposure of the activation marker P-selectin in platelets exposed to different concentrations of EHEC. Consistent with the aggregometry data (Figure 2A), platelet exposure to different EHEC concentrations did not result in platelet integrin α_IIb_β_3_ activation (Figure 2B) or the surface exposure of P-selectin (Figure 2C).

To exclude the possibility that EHEC materials prevent or stimulate agonist-induced platelet aggregation, we next compared EHEC solution in PRP that was activated with the platelet agonist adenosine diphosphate (ADP), which promotes human platelet aggregation via the G protein-coupled receptor P2Y12. Importantly, in the dynamic range of ADP concentration, we did not detect an altered ADP-triggered aggregation response when EHEC was present in PRP (Figure 3A). We also did not detect differences in platelet response after incubation with different EHEC concentrations when stimulated with a threshold concentration of ADP that resulted in secondary platelet activation (Figure 3B). We applied the same experimental setup in flow cytometry to test whether the exposure of the platelets to different EHEC concentrations would lead to increased ADP-stimulated α_IIb_β_3_ activation (Figure 3C) and surface exposure of the activation marker P-selectin (Figure 3D). Collectively, our results demonstrate that EHEC exposure had no effect on ADP-stimulated platelet activation.

To test whether the biocompatibility of EHEC solution was similar to materials already in clinical use as plasma expander solutions (Gelfusal^®^, Volulyte^®^, Vitafusal^®^), we compared the EHEC solution to these solutions in the PRP of the same donor. Indeed, the EHEC solution was comparable to Gelfusal^®^ and Vitafusal^®^ and, as expected, EHEC showed a reduced platelet aggregation response at resting conditions relative to Volulyte^®^ (Figure 4A) [24]. Furthermore, when comparing the potential role of EHEC to other clinically applicable materials (i.e., Gelfusal^®^, Volulyte^®^, Vitafusal^®^) in ADP-activated PRP, there was neither an impairment nor an enhancement of ADP-induced platelet aggregation (Figure 4B). Altogether, these data imply that EHEC does not interfere with the physiological platelet aggregation response of the hemostatic system.

As EHEC did not interfere with platelet aggregation ex vivo, our data argue for the applicability of EHEC solution as a biocompatible material for human blood plasma.

## 3. Discussion

Rotational thromboelastometry experiments showed that the supplementation of blood with different EHEC concentrations does not enhance ex vivo clot formation. This is in contrast to gelatin-based plasma expanders, which lead to hypercoagulability [24]. Based on platelet aggregometry and flow cytometry analyses with PRP, we conclude that EHEC does not significantly activate or interfere with platelet function, which is an important requirement for its potential therapeutic application in the circulation. This is in contrast to existing polymer-based nanomaterials used in blood, which may have unfavorable properties that affect the clotting system, especially in long-term applications, therefore virtually precluding their use as nanocarriers [25,26]. For instance, HES has been described to enhance fibrinolysis by diminishing the inhibition of plasmin by its serpin inhibitor α2-antiplasmin [27]. In an ex vivo whole blood thromboelastometry analysis, HES was demonstrated to have an anticoagulant effect depending on the dilution ratio [25], which may be due to a decay in factor VIII levels [25]. Moreover, HES was shown to impair collagen and epinephrine-induced platelet aggregation in PRP [28]. In clinical use, HES infusion was found to cause a slight but significant decrease in platelet count.

Gelatin-based plasma substitutes also have adverse effects on the clotting system. Gelatin-based colloidal solutions reduce the clotting capacity of fresh blood, resulting in clots with decreased weight, mean shear modulus, and increased bleeding times [29,30]. The analysis of these clots with scanning electron microscopy revealed a less extensive fibrin mesh [29]. Our analyses also show increased clotting times and clot formation times, but only at the highest EHEC concentrations (10% *v*/*v*). In addition to the effects of gelatin on the coagulation cascade, velocity and ristocetin-induced platelet agglutination were decreased and thrombin–antithrombin complexes were reduced [30]. Taken together, this highlights the need for new nanomaterials such as EHEC that are inert to platelet function, i.e., platelet aggregation.

As shown by shear analysis, another favorable property of EHEC nanomaterials for use as biocompatible carrier polymers in the blood circulation, which clearly distinguishes this material from other biocompatible polymers, is that the rheological properties of EHEC solutions resemble the non-Newtonian rheological profile of human blood. This could enable its use in various medical and analytical applications that require blood-borne particles. The topical application of EHEC has previously been proposed as a potential delivery system for local anesthetics [31]. Our results suggest that intravascular in vivo applications of EHEC-based materials should be investigated, and may turn out to be useful for targeted chemotherapy approaches to combat tumor growth.

## 4. Materials and Methods

### 4.1. Ethyl Hydroxyethyl Cellulose (EHEC)

Ethyl hydroxylethyl cellulose (EHEC)—the commercial product BERMOCOLL EHM 200 (Akzo Nobel Functional Chemicals AB, Stenungsund, Sweden)—was fractionated by means of a liquid–liquid phase separation. Water was used as solvent, and acetone was used as non-solvent. At the end, the EHEC had an apparent weight average molar mass Mn* of 14,700 g/moL, an apparent weight average molar mass Mw* of 492,000 g/mol, and a dispersity Đ of 33, analyzed by means of size exclusion chromatography (SEC). To prepare the final EHEC solution, the fractionated polymer was diluted with a concentration of 0.38 % (*w*/*w*) in isotonic NaCl solution. The fractionation and preparation were performed at the WEE-Solve GmbH.

### 4.2. Size-Exclusion Chromatography (SEC)

The molecular mass distribution of EHEC was determined by means of size-exclusion chromatography (SEC) using a refractive index detector (RI-71, Shodex, Munich, Germany). The columns HEMA Bio 40, HEMA Bio 1000, and Suprema 3000 (PSS Polymer Standards GmbH, Mainz, Germany) were used, and the determination was performed in salt solution (0.05 mol NaHCO_3_ and 0.1 mol NaNO_3_ in water) as eluent. The calibration curve was performed with dextran standards (PSS Polymer Standards GmbH, Mainz, Germany). The analysis was performed at the WEE-Solve GmbH.

### 4.3. Rheology Studies

Flow curves were determined by means of an air-bearing rotational rheometer UDS 200 (Anton Paar, Graz, Austria) in combination with a Paar Physica Viscotherm V2 thermostat. Z1 cylinder geometry (double gab) was used. 22 mL of the sample was filled inside the lower cylinder without filtration. The upper cylinder was lowered into the measuring position, surplus material was removed, the samples were allowed to equilibrate to 37 °C for 5 min, and the measurement was started. Measurement profile: shear rate decreasing from 300 to 10 1/s (for volulyte 6%, gelafusal 4%, and vitafusal 6%, respectively) and 1000 to 10 1/s (for human blood and EHEC) (5 data points per decade; measuring time: logarithmic increasing from 1 to 30 s).

### 4.4. Blood Collection

Whole blood anticoagulated with tri-sodium citrate (109 mM, 1:10) from healthy donors was collected from the antecubital area after informed consent was retrieved according to our institutional guidelines and the Declaration of Helsinki.

### 4.5. Rotational Thromboelastometry (ROTEM)

For the reconstitution of the clotting by the extrinsic and the intrinsic pathway, 300 μL whole blood was added to single-use reagent EX-TEM S or IN-TEM S according to the manufacturer’s instructions (Werfen GmbH, Munich, Germany). For monitoring the effect of EHEC on the system, different amounts of 0.38% *w*/*w* EHEC solution were added to 300 μl whole blood to obtain final concentrations of 96.25 μg/mL (5% *v*/*v*), 192.5 μg/mL (10% *v*/*v*), and 385 μg/mL (20% *v*/*v*). Additionally equal volumes of isotonic NaCl solution (154 mM) were used as vehicles. Clotting time and clot formation time were measured by a whole blood hemostasis analyzer (ROTEM delta, Tem GmbH, Munich, Germany).

### 4.6. Aggregometry

Whole blood was centrifuged at 200× *g* for 10 min at room temperature (RT). The platelet-rich plasma (PRP) was collected and adjusted to 2 × 108 cells/mL with HEPES buffer (150 mM NaCl, 5 mM KCl, 1 mM MgCl 2, 10 mM D-glucose, 10 mM HEPES, pH 7.4). Then, 500 μL of PRP were centrifuged at 2000× *g* for 10 min at RT to acquire platelet-poor plasma, which was used as a reference. To examine the nanomaterial effect in PRP, 20 μL of each were added to 180 µL PRP, corresponding to final concentrations of 385 μg/mL EHEC, 6 mg/mL Volulyte, 4 mg/mL Gelafusal, and 6 mg/mL Vitafusal. Equal volumes of isotonic NaCl solution (154 mM) were used as vehicles. Where indicated, platelets were stimulated with 5 µM ADP to achieve an 80% aggregation response (Hart biologicals, Hartlepool, UK). The platelet aggregation was quantified by light transmission aggregometry using an APACT 4S Plus aggregometer (Diasys Greiner, Holzheim, Germany).

### 4.7. Flow Cytometry

Platelet-rich plasma (PRP) was diluted 1:5 with HEPES buffer (150 mM NaCl, 5 mM KCl, 1 mM MgCl 2, 10 mM D-glucose, 10 mM HEPES, pH 7.4). To investigate the effect of the EHEC in PRP, different amounts of 0.38% *w*/*w* EHEC solution were added to diluted PRP to obtain final concentrations of 96.25 μg/mL, 192.5 μg/mL, and 385 μg/mL. The samples were incubated with 5μM ADP (Merck KGaA, Darmstadt, Germany) for 5 min at room temperature. To examine the influence of the nanomaterial in PRP without activation by ADP, mixtures of diluted PRP with 0.38% *w*/*w* EHEC solution were prepared in the same manner without activation by ADP. Activated platelets and non-activated PRP/EHEC mixtures were stained with APC (allophycocyanin)-labeled mouse anti-human CD62P (P-Selectin) (BD Biosciences, San Jose, CA, USA) with FITC (fluorescein isothiocyanate)-labeled mouse anti-human PAC-1 (BD Biosciences, San Jose, CA, USA) for 15 min at room temperature. Then, 1ml HEPES buffer was added to the samples and the samples were analyzed immediately on the flow cytometer. Per sample, at least 10,000 events were collected. As controls, unstimulated samples were used from the same donors. For the study, a BD FACSCanto II flow cytometer with BD FACSDiva software (v6.1.3, BD Biosciences, Heidelberg, Germany) was applied. The histograms were generated using FlowJo V10 software (FlowJo LLC, Ashland, OR, USA).

## 5. Patents

EP000003072502A1, 25.03.15, “Rheological blood replacement solution and uses thereof”.

## Figures and Tables

**Figure 1 ijms-23-06432-f001:**
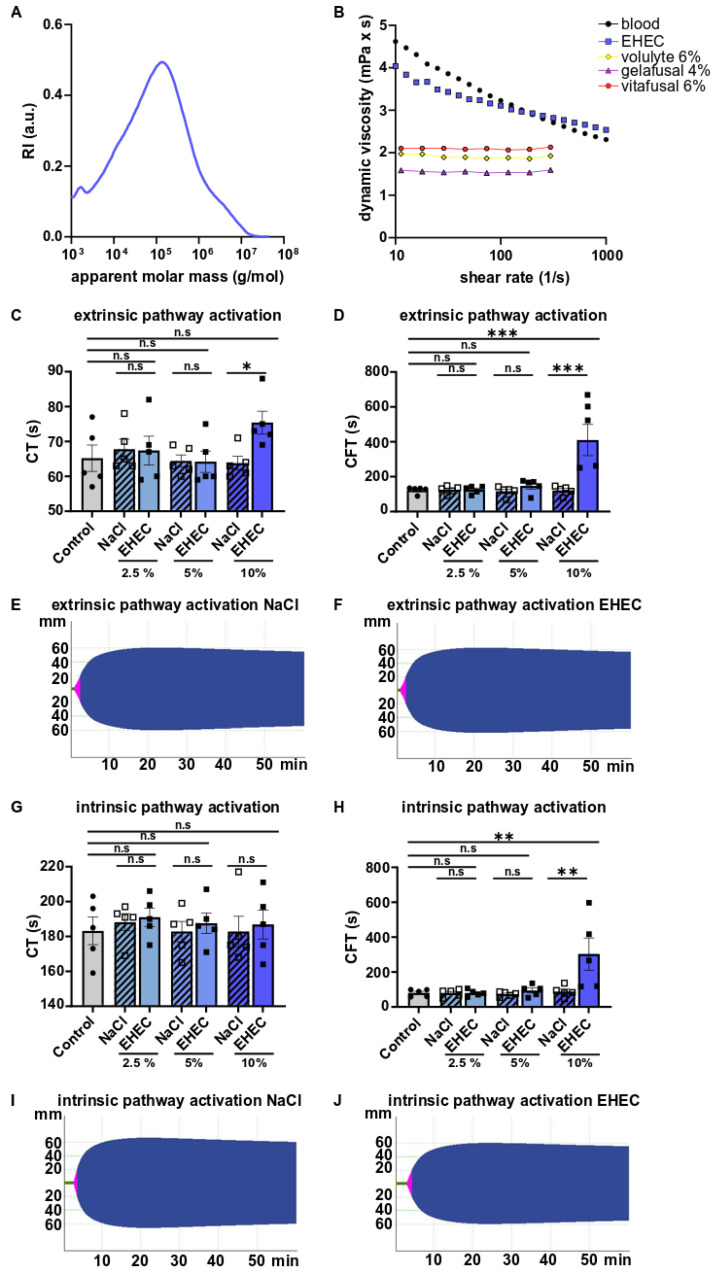
Characterization of ethyl hydroxyethyl cellulose. (**A**) Molar mass distribution. (**B**) Flow curves of blood (black) and the different biocompatibles: EHEC (blue), Volulyte (yellow), Gelafusal (purple), and Vitafusal (red). (**C**) Clotting time and (**D**) clot formation time of whole blood and whole blood diluted with different NaCl (vehicle) and EHEC concentrations (*n* = 5). Coagulation was initiated by tissue thromboplastin. (**E**,**F**). represent rotational thromboelastometry (ROTEM) graphs of whole blood diluted with 5% NaCl (**E**) and 5% EHEC (**F**) activated by tissue thromboplastin (EXTEM). Clotting time appears in green and clot formation time in pink. (**G**) Clotting time and (**H**) clot formation time of whole blood and whole blood diluted with different NaCl (vehicle) and EHEC concentrations (2.5%, 5% and 10%) (*n* = 5). Coagulation was initiated by ellagic acid (INTEM). (**I**,**J**) represent rotational thromboelastometry (ROTEM) graphs of whole blood diluted with 5% NaCl (**I**) and EHEC (**J**) activated by ellagic acid (INTEM). Clotting time appears in green and clot formation time in pink. Black circle: whole blood, white square: whole blood+NaCl, black square: whole blood+EHEC. All data are expressed as the means ± SEM. Statistical comparisons were performed using a one-way ANOVA. * *p* < 0.05, ** *p* < 0.01, *** *p* < 0.001, n.s not significant.

**Figure 2 ijms-23-06432-f002:**
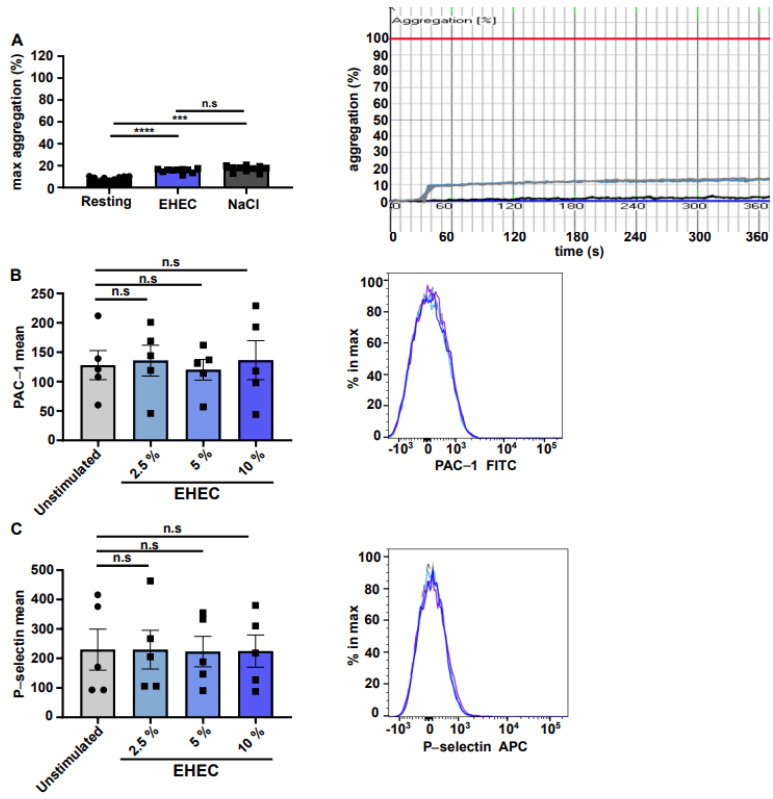
The effect of ethyl hydroxyethyl cellulose (EHEC) on platelet function under resting conditions. (**A**) Aggregometry histograms and representative curves of resting platelets (black) and platelets incubated with EHEC (blue) or NaCl (grey) (*n* = 10). Mean fluorescence intensity of (**B**) FITC-conjugated PAC-1 antibody and (**C**) APC-conjugated P-selectin antibody of unstimulated platelets or platelets exposed to different EHEC concentrations (the shades represent the concentration increase) (*n* = 5). All data are expressed as the means ± SEM. Statistical comparisons were performed using a one-way ANOVA. *** *p* < 0.001, **** *p* < 0.0001, n.s not significant.

**Figure 3 ijms-23-06432-f003:**
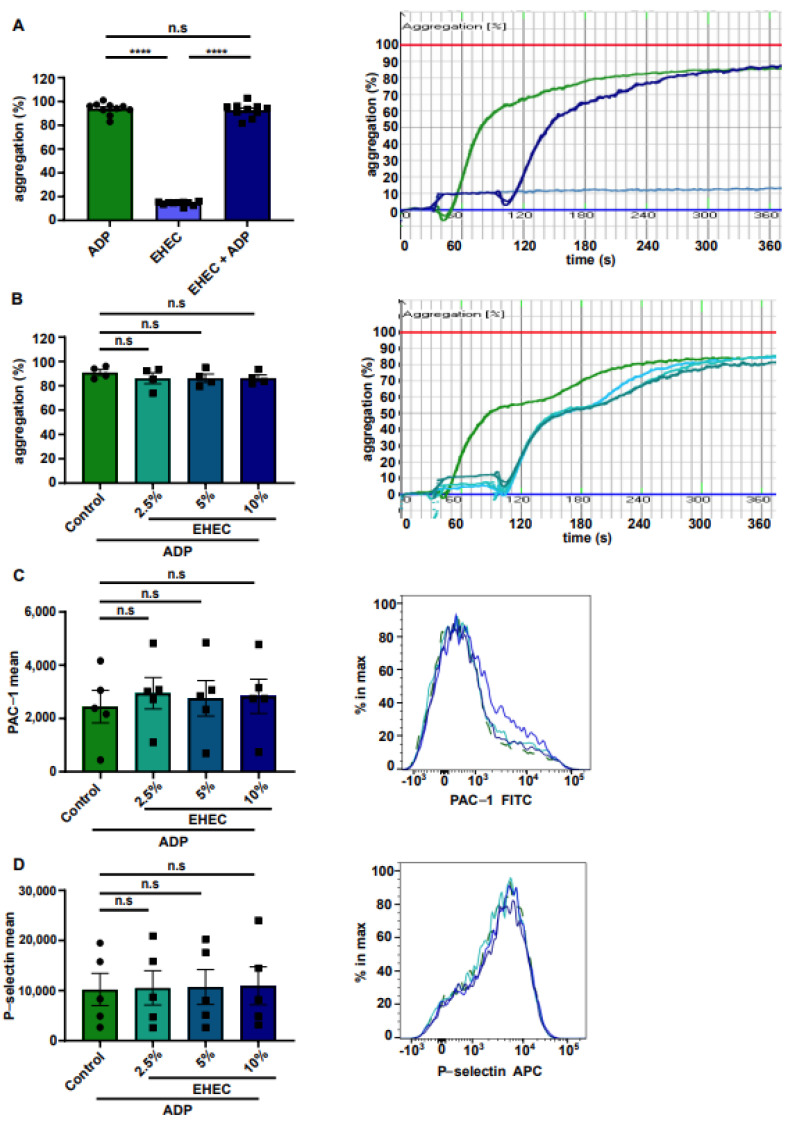
The effect of ethyl hydroxyethyl cellulose (EHEC) on ADP-stimulated platelets. (**A**) Aggregometry histograms and representative curves of ADP-stimulated platelets with high ADP concentrations (green), resting platelets incubated with EHEC (blue) and ADP-stimulated platelets incubated with EHEC (dark blue) (*n* = 10). (**B**) Aggregometry histograms and representative curves of ADP-stimulated platelets with threshold ADP concentrations (green) and ADP-stimulated platelets incubated with different EHEC concentrations (the shades represent the concentration increase). (**C**) Mean fluorescence intensity of the FITC-conjugated PAC-1 antibody and (**D**) APC-conjugated P-selectin antibody of ADP-stimulated platelets incubated without or with different EHEC concentrations (the shades represent the concentration increase) (*n* = 5). All data are expressed as means ± SEM. Statistical comparisons were performed using a one-way ANOVA. **** *p* < 0.0001, n.s not significant.

**Figure 4 ijms-23-06432-f004:**
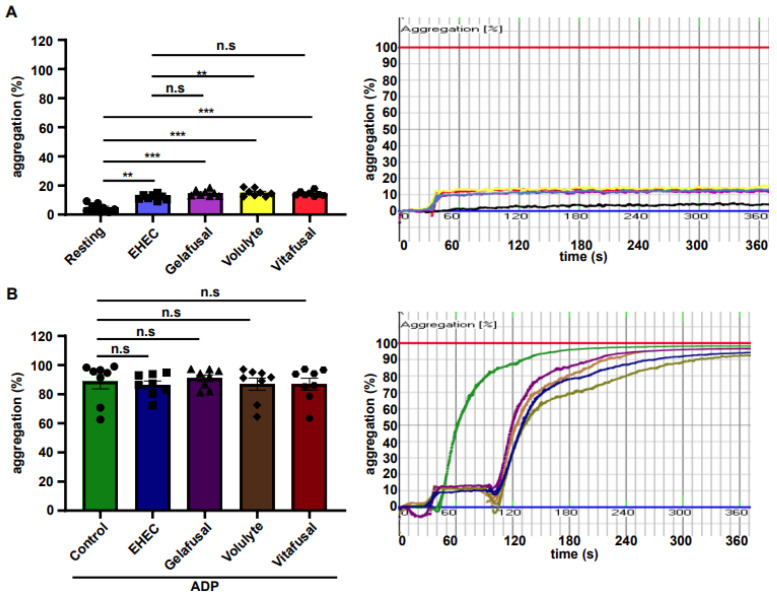
Comparison between ethyl hydroxyethyl cellulose (EHEC) and commercial plasma expander solutions on platelet aggregation. Aggregometry histograms and representative curves of (**A**) resting platelets (black) and platelets incubated with EHEC (blue), Gelafusal (purple), Volulyte (yellow), and Vitafusal (red) (*n* = 10). (**B**) ADP-stimulated platelets (green) and platelets stimulated with ADP and exposed to EHEC (dark blue), Gelafusal (dark purple), Volulyte (brown) or Vitafusal (dark red) (*n* = 10). All data are expressed as the means ± SEM. Statistical comparisons were performed using the one-way ANOVA. ** *p* < 0.01, *** *p* < 0.001, n.s not significant.

**Table 1 ijms-23-06432-t001:** Composition of various plasma expander solutions.

Gelafusal^®^		Volulyte^®^		Vitafusal^®^	
w (g) in 1000 mL	Compound	w (g) in 1000 mL	Compound	w (g) in 1000 mL	Compound
40.000	Gelatine polysuccinate	60.000	Hydroxyethyl starch	60.000	Poly(O-2-hydroxyethyl) starch
3.675	Sodium acetate trihydrate	4.630	Sodium acetate trihydrate	3.700	Sodium acetate trihydrate
4.590	Sodium chloride	6.020	Sodium chloride	6.000	Sodium chloride
0.403	Potassium chloride	0.300	Potassium chloride	0.400	Potassium chloride
0.133	Calcium chloride dihydrate	0.300	Magnesium chloride hexahydrate	0.134	Calcium chloride dihydrate
0.203	Magnesium chloride hexahydrate			0.200	Magnesium chloride hexahydrate

## Data Availability

Not applicable.

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
