# Peer review of "Ethyl Hydroxyethyl Cellulose—A Biocompatible Polymer Carrier in Blood"

_ijms, 2022, doi:10.3390/ijms23126432_

Round 1

Reviewer 1 Report

The work entitled “Ethylhydroxyethyl cellulose – A Biocompatible Polymer Carrier in Blood” evaluates the potentiality of EHEC to induce ex vivo clot formation, platelet aggregation, activation by thromboelastometry, aggregometry, and flow cytometry. EHEC was compared with commercial products and demonstrated that it may be suitable for the use as a biocompatible carrier material in the blood circulation and for applications in flow-dependent diagnostics.

The introduction of the subject is very clearly done. However, more emphasis should be given to the novelty; in fact, that should be highlighted. This and the presence of small English mistakes along the manuscript are the only alterations the manuscript requires.

The obtained data is very well described and discussed. It is scientifically sound and one important thing the authors confronted the information with the literature and discussed the results as a whole and not as distinctive pieces. The work is very pertinent and it was a very easy reading. I recommend its publication after minor revision.

Reviewer 2 Report

This manuscript is focused on the evaluation of ethyl hydroxyethylcellulose (EHEC) as a polymer carrier in blood. Generally, the paper is well written and appears scientifically competent within my area of expertise. In my opinion, the manuscript can be considered for publication in the IJMS after addressing the following comments:

  1. The EHEC sample is not chemically characterized in any way. At least the degree of substitution of cellulose by ethyl and hydroxyethyl groups must be given. The degree of substitution with functional groups, the molecular weight and the dispersity are key parameters that affect the physicochemical properties of the polymer, including viscosity, aggregation ability, solubility, etc.
  2. It would be of great benefit to characterize the EHEC solution by dynamic light scattering for the presence of polymer unimers and associates.
  3. Figure 1A and across the text: The molecular weight (in contrast to the molar mass) is unitless. Either remove g/mol or use term molar mass across the text.
  4. Section 4.1: Please double check the molar mass values. Either Mw or Mn is 10 times higher/lower than estimated based on the dispersion of 3.3.
  5. Section 4.1: ‘non-uniformity’ is a conversational term, please use the term ‘dispersity’ (Đ, crossed D).

Round 2

Reviewer 2 Report

The authors have successfully addressed most of the reviewer's comments. In my opinion  the manuscript can be accepted for publication in IJMS.